# Insights to the Structural Basis for the Stereospecificity of the *Escherichia coli* Phytase, AppA

**DOI:** 10.3390/ijms23116346

**Published:** 2022-06-06

**Authors:** Isabella M. Acquistapace, Emma J. Thompson, Imke Kühn, Mike R. Bedford, Charles A. Brearley, Andrew M. Hemmings

**Affiliations:** 1School of Biological Sciences, University of East Anglia, Norwich Research Park, Norwich NR4 7TJ, UK; isabella.acquistapace@iit.it (I.M.A.); e.thompson1@uea.ac.uk (E.J.T.); c.brearley@uea.ac.uk (C.A.B.); 2AB Vista, Feldbergstrasse, 64293 Darmstadt, Germany; imke.kuehn@abvista.com; 3AB Vista, Blenheim Road, Marlborough SN8 4AN, UK; mike.bedford@abvista.com; 4School of Chemistry, University of East Anglia, Norwich Research Park, Norwich NR4 7TJ, UK; 5College of Food Science and Technology, Shanghai Ocean University, Shanghai 201306, China

**Keywords:** phytase, AppA, enzymology, catalytic mechanism, stereospecificity, enzyme structure, X-ray crystallography

## Abstract

AppA, the *Escherichia coli* periplasmic phytase of clade 2 of the histidine phosphatase (HP2) family, has been well-characterized and successfully engineered for use as an animal feed supplement. AppA is a 1D-6-phytase and highly stereospecific but transiently accumulates 1D-*myo*-Ins(2,3,4,5)P_4_ and other lower phosphorylated intermediates. If this bottleneck in liberation of orthophosphate is to be obviated through protein engineering, an explanation of its rather rigid preference for the initial site and subsequent cleavage of phytic acid is required. To help explain this behaviour, the role of the catalytic proton donor residue in determining AppA stereospecificity was investigated. Four variants were generated by site-directed mutagenesis of the active site HDT amino acid sequence motif containing the catalytic proton donor, D304. The identity and position of the prospective proton donor residue was found to strongly influence stereospecificity. While the wild-type enzyme has a strong preference for 1D-6-phytase activity, a marked reduction in stereospecificity was observed for a D304E variant, while a proton donor-less mutant (D304A) displayed exclusive 1D-1/3-phytase activity. High-resolution X-ray crystal structures of complexes of the mutants with a non-hydrolysable substrate analogue inhibitor point to a crucial role played by D304 in stereospecificity by influencing the size and polarity of specificity pockets A and B. Taken together, these results provide the first evidence for the involvement of the proton donor residue in determining the stereospecificity of HP2 phytases and prepares the ground for structure-informed engineering studies targeting the production of animal feed enzymes capable of the efficient and complete dephosphorylation of dietary phytic acid.

## 1. Introduction

Phosphorus is essential for life and one of the most important minerals for animal nutrition. Phytic acid (*myo*-Inositol hexakisphosphate (InsP_6_)) is the major storage form of phosphorus present in the plant-derived constituents of common animal feeds [1] and therefore key to animal growth. Paradoxically, as it passes through the gut, phytic acid binds to minerals such as iron, zinc, and calcium reducing their adsorption in the intestine. As a consequence, phytic acid is also recognized as a potent antinutrient [2]. Phytases are phosphomonoesterases which catalyse the dephosphorylation of phytate releasing orthophosphate. Animals rely on phytases produced by their commensal microbiota, yet in monogastrics such as pigs and poultry, the capacity of these phytases to break down dietary phytic acid is limited. A mucose-bound phytase from the small intestine of chickens was reported but this is positioned in the wrong part of the gut to be meaningfully effective [3]. Undigested or partially digested phytic acid is excreted and subsequently broken down by soil phytases, which can lead to eutrophication of freshwater lakes and estuarine waters [4,5].

To increase the efficiency of the conversion of phytic acid into available dietary phosphate and thus reduce faecal inositol polyphosphate levels, non-ruminant animal feeds are routinely supplemented with exogenous phytases. Of the major classes of phytase, those belonging to clade 2 of the histidine phosphatase superfamily (HP2) have found the most widespread use. This is due to their high specific activity towards phytate, absence of cofactor dependence, and pH optima in the acid regime. Notable examples are engineered variants of the *Escherichia coli* periplasmic phytase, AppA, and secreted fungal phytases such as PhyA of *Aspergillus niger* [6,7]. This area continues to attract interest with numerous reports of new enzymes uncovered through bioprospecting and of variants with high activity, tailored specificity or enhanced stability to environmental assault, the need for the latter arising both from the elevated temperatures to which these enzymes are exposed during the feed pelleting process and from proteinases of the animal digestive system [8,9,10].

The crystal structures of a number of HP2 phytases (HP2P) have been solved from a variety of both bacterial [11,12,13,14,15] and fungal [16,17,18] sources revealing a characteristic tertiary structure consisting of two structural domains: an *α*- and an *α*/*β*-domain separating the active site. The fold of the *α*/*β*-domain is well-conserved in HP2P, while the *α*-domain is subject to enzyme-specific changes [19]. HP2Ps catalyse the hydrolysis of phytate via an obligatory phosphohistidine intermediate (Figure 1) [20].

Two amino acid sequence motifs central to catalysis are located at the base of this active site cleft on polypeptide loops connecting the two domains. The first of these possesses a consensus RHGxRxh sequence (where h represents a hydrophobic amino acid). This motif is involved in substrate binding and contains the eponymous nucleophilic histidine observed in all HP2 family members [20,21,22,23]. The catalytic proton donor is found in a second, short, conserved motif with sequence HD positioned such that the aspartic acid residue functions as the consensus proton donor in the catalytic mechanism [11,23,24,25,26]. The proton donor is required for formation of a phosphohistidine intermediate and release of the lower inositol phosphorylated product of hydrolysis. The same residue then acts as a general base catalysing regeneration of the catalytic histidine through loss of orthophosphate. HP2Ps can be grouped according to their stereospecificities, i.e., the specific position of the phosphate ester group of the substrate at which hydrolysis is initiated. The *E. Coli* periplasmic phytase, AppA, is a 1D-6-phytase (EC 3.1.3.2) and initial hydrolysis preferentially liberates the phosphate at the 1D-6-position on the inositol ring [27]. Stereospecific HP2Ps such as AppA are highly active and act stepwise on phytate to generate a characteristic progression of inositol polyphosphates. While the initial steps of hydrolysis by AppA are rapid, the efficiency of hydrolysis of the major InsP_4_ intermediate, 1D-*myo*-Ins(2,3,4,5)P_4_, is reduced leading to its accumulation [28,29,30]. This is also true of the lower inositol polyphosphates produced subsequently by further hydrolysis. Recognition of this deficiency led to efforts to evolve AppA to generate variants which show more efficient hydrolysis of InsP_4_ and InsP_3_ [31].

The multiple inositol polyphosphate phosphatases (MINPPs) constitute a distinct evolutionary group within clade 2 of the histidine phosphatase superfamily [24,32]. The function of many MINPPs is uncertain despite evidence of the role of these enzymes in a variety of cellular processes and organisms [33,34,35,36]. Similar to other HP2P enzymes, MINPPs carry the RHGxRxh sequence motif involved in substrate binding and catalysis. However, instead of an HD proton donor motif, the residues of an amino acid triplet, frequently with the sequence HAE, are presumed to provide an equivalent function [25]. MINPPs were so-named because of their broad substrate specificity compared with other HP2Ps. However, unlike other HP2P enzymes, MINPPs lack a strong stereospecificity towards phytic acid, producing instead a variety of InsP_5_s and lower inositol polyphosphates [25,37,38,39,40]. This led to the suggestion that MINPPs may provide a route to efficient dephosphorylation of dietary phytate which avoids the accumulation of lower inositol polyphosphates observed with animal feed enzymes based on AppA [40].

To help explain the factors determining the stereospecificity of *E. coli* AppA, the role of the identity and position of the catalytic proton donor was investigated. Four AppA active site variants were generated by site-directed mutagenesis of the active site HDT amino acid sequence motif containing the catalytic proton donor, D304, to introduce stepwise the HAE motif commonly found in MINPPs. Kinetic analysis of active site variants was supplemented with analysis of high-resolution X-ray crystal structures of active site variants bound to a non-hydrolysable substrate analogue at pH 3.0. Furthermore, additional crystal structures of phosphate-bound, and phosphohistidine intermediate forms of the enzyme were determined providing snapshots along the complete structural catalytic cycle of the enzyme. Our results provide new insights to catalysis and the stereospecificity of this important animal feed enzyme and provide a basis for future attempts to generate variants capable of the rapid, exhaustive hydrolysis of phytate.

## 2. Results

### 2.1. AppA Variants Can Be Efficiently Expressed in the E. coli Cytoplasm Using a DsbC-Producing Strain

N-terminal 6xHis-tagged AppA (6H3C-AppA-HDT) and four proton donor motif variants D304A (6H3C-AppA-HAT), D304E (6H3C-AppA-HET), D304A/T305E (6H3C-AppA-HAE), and T305E (6H3C-AppA-HDE) were expressed in the cytoplasm of the *E. coli* expression strain, Shuffle Express T7, and purified by immobilized metal affinity chromatography. The hexahistidine affinity tag was subsequently removed from each protein with 3C protease and the recombinant enzymes purified to homogeneity. The fold of AppA is characterized by three consecutive disulfide bridges and an additional fourth C-terminal non-consecutive disulfide, the formation of which is catalysed by the periplasmic disulfide isomerase DsbC [41]. Shuffle Express strains are engineered to constitutively express DsbC in their cytoplasm and have been used previously to produce recombinant AppA [42]. Following purification and removal of the N-terminal hexahistidine affinity tag, this method provided structural biology grade recombinant enzymes for subsequent enzymatic and structural analyses. The absence of native *E. coli* periplasmic AppA from the expression strain as a contaminant of the purified recombinant enzymes by was verified by the absence of phytase activity following the expression driven from an empty expression vector (data not shown).

### 2.2. Crystal Structure of the Complex with a Phytate Analogue Inhibitor Allows Identification of Specificity Pocket Residues

AppA is a 1D-6-phytase [43]. To gain an understanding of the structural basis for the primary stereospecificity of the wild-type enzyme our first objective was to identify specific interactions determining the recognition of phytic acid. While a crystal structure is available of the complex of an inactive AppA-H17A variant with InsP_6_ [11], this cannot faithfully represent the interactions of the wild-type enzyme. Indeed, the substrate binds so as to insert the 1D-3-phosphate in the catalytic pocket despite the pronounced 1D-6-phytase activity of the enzyme. In many cases, *myo*-inositol hexakissulfate (InsS_6_) is reported to act as a competitive inhibitor of phytases and in crystal structures is assumed to mimic the substrate by adopting a pseudo-productive binding mode [25,40,44,45]. Attempts were therefore made to solve the crystal structure of InsS_6_-inhibited AppA by soaking crystals grown at pH 6.5; however, no resolvable electron density for the bound inhibitor was observed. To investigate this observation, we carried out phytase inhibition assays at pH 3.0 and 5.5, the latter being the consensus pH for phosphate release assays involving HP2 phytases. This revealed InsS_6_ to inhibit the enzyme with an IC_50_ of ~10 μM at pH 3.0 but ~1 mM at pH 5.5 (Appendix A). Through subsequent experimentation we were able to uncover crystal soaking parameters leading to a high-resolution structure of the complex of wild-type AppA and the inhibitor at pH 3.0. Importantly, this delivered structural data at a pH at which the enzyme is maximally active. Solved by molecular replacement using the structure of apo-AppA (PDB entry 1DKL), InsS_6_ was modelled to the experimental ligand OMIT electron-density maps (Appendix A) with the sulfomonoester group at position 1D-6 of the inositol ring pointing towards the catalytic histidine residue, His17. This assignment was unambiguous due to clear electron density for the unique axial 2-sulfate. The final structure, refined using data of 1.72 Å resolution provided a model with low *R*_free_ (20.7%) (Appendix A). Comparison of this structure with that of the H17A mutant in complex with phytate (PDB entry 1DKQ) revealed no significant conformational change in the overall protein structure (rmsd 0.31 Å over 355 Cα atoms), although the InsS_6_ inhibitor was displaced from the position adopted by phytate by an average of 1.1 Å over the six carbons of the inositol ring. Despite the shift in position of the inositol ring and change in orientation of binding of the ligand, conformational changes in the active site were small and restricted to individual amino acid sidechains. The sole notable change is the static disorder modelled for the sidechain of the proton donor residue, D304. This sidechain has one conformer pointing towards the bridging oxygen of the 6-sulfomonoester group bound in the catalytic pocket at a distance of 2.9 Å and held in place by interaction with the guanidinyl group of R16, while the other conformer points away. The interactions with the pseudosubstrate inhibitor in the complex are also similar to those described for other HP2 1D-6-phytases. For example, an rmsd 0.52 Å was observed for the *Hafnia alvei* phytase (PDB entry 4ARO) over 302 aligned residues.

Subject to the caveat that the p*K*_a_s of the sulfate groups of InsS_6_ differ from those of the phosphates of phytic acid, the structure of the complex of InsS_6_ with wild-type AppA provides a reasonable basis for identification of enzyme–substrate interactions. Specificity pockets for the binding of each of the six phosphates of InsP_6_ were thus inferred by identifying all amino acids within 5 Å of the corresponding sulfate group of the ligand. In this scheme, the residues of pocket A bind the scissile phosphate group and represent the catalytic centre. As previously described [11], this pocket is symmetrical in structure and charge, presenting guanidino groups from two arginine residues (R20 and R92) on each side of the phosphate. From a vantage point positioned behind the inositol ring and looking through it towards the A-subsite and the guanidinyl groups of R20 to the right and R92 to the left, the remaining specificity subsites are then labelled B–F in a counter clockwise fashion, following the order of decreasing sulfate number attached to the *myo*-inositol ring (Figure 2).

Non-symmetric substitution of the ring, or removal of phosphate (sulfate) from symmetric substrate (analog), necessitates assignment by 1D- or 1L- notation; we used 1D-, throughout. It is likely that the subsites we define are representative of the subsites involved in recognition and binding of InsP_6_. Of note is the fact that specificity pockets A and B are the most highly populated by active site residues while pockets D and E are the least (Appendix A). This trend is broadly characteristic of all HP2 phytases for which crystal structures are available. However, we note that specificity subsites D and E are more highly populated in the MINPP from *Bifidobacterium longum* where residues of an unusual polypeptide insertion in the α-domain contribute to additional interactions with the substrate [40].

### 2.3. Motif Variants Map the Influence of the Proton Donor on Catalysis

Four AppA active site variants were generated by site-directed mutagenesis of the active site HDT amino acid sequence motif containing the presumed catalytic proton donor, D304, to introduce stepwise the HAE motif commonly found in MINPPs. To assess the effect of varying the identity and position of the presumed catalytic proton donor residue, phosphate release assays were performed to allow estimation of the Michaelis–Menten kinetic parameters, *K*_m_ and *k*_cat_ (Table 1).

These measurements were based on initial rate estimates derived subject to the condition that less than 10% of the total available phosphate in the substrate had been hydrolysed during the assay. Comparable *K*_M_ values were obtained for the wild-type enzyme and AppA-HET, while the remaining mutants displayed lower Michaelis constants. Notably, *K*_M_ was 5-fold lower than wild type for the proton donor-less AppA-HAT variant and the MINPP-like AppA-HAE double mutant. The low turnover numbers (*k*_cat_) for these hydrolysis-compromised variants (only ~0.1% of that seen for the wild-type enzyme) suggests that breakdown of the phosphohistidine intermediate is rate limiting. Under the rapid equilibrium assumption, we assumed that the rate at which the equilibrium is established for substrate binding is fast compared with the rate at which the ES complex forms product. Hence, their *K*_M_ values may approximate to the dissociation constants (*K*_d_) for binding of phytate to an active site which lacks an identifiable proton donor residue at position 304. *K*_cat_ for the AppA-HET variant is roughly ten times greater than for these proton-donor-less variants, but it has a *K*_M_ essentially unaltered from the wild-type enzyme pointing not only towards a role for a carboxylate group in the energetics of ligand binding but also indicating that glutamate cannot directly functionally substitute for aspartate at residue position 304. *K*_M_ also decreased for the AppA-HDE mutant while its turnover number is reduced to ~17% of the wild-type enzyme.

### 2.4. The Position and Identity of the Nominal Proton Donor Residue Influences Stereospecificity

To evaluate the stereospecificity of hydrolysis of InsP_6_ by wild-type AppA and its variants, the products of limited hydrolysis were separated by HPLC. Enzyme concentrations and the durations of reaction were adjusted to halt hydrolysis at comparable stages. Specifically, the percentage of depletion of InsP_6_ or the percentage of formation of InsP_1_/released phosphate was held constant for each mutant in order to compare systematically the HPLC elution profiles of hydrolysis intermediates. Note, however, that the HPLC method employed cannot distinguish between enantiomers which coelute. Thus, integration of peak areas in the elution profiles allowed not identification of absolute stereochemistry but estimation of the relative proportions of 1D- and or 1L-*myo*-Ins(1,2,3,4,5)P_5_ (hereafter InsP_5_ [4/6-OH] noting that the enantiomers are not resolvable) and 1D- and or 1L-*myo*-Ins(1,2,4,5,6)P_5_ (hereafter InsP_5_ [1/3-OH]) generated by action of the enzyme variants on InsP_6_.

Assuming that the well-characterized pathway of sequential hydrolysis of phytate [43,46] holds for our preparation of AppA, hydrolysis leads initially to the generation of one predominant InsP_5_ species, presumably 1D-*myo*-Ins(1,2,3,4,5)P_5_ (hereafter referred to as InsP_5_ [6-OH]). Further hydrolysis yields 1D-*myo*-Ins(2,3,4,5)P_4_ as the predominant InsP_4_ intermediate. Plotting the relative amounts of inositol polyphosphate hydrolysis intermediates detected as a function of time (Figure 3) revealed that initial hydrolysis of phytate by AppA leads to almost complete conversion to InsP_5_ before breakdown of this intermediate begins. When InsP_5_ is processed to InsP_4_, the latter becomes a substrate. This is again in keeping with previous observations [27,46,47].

By integration of peak areas in the HPLC elution profile of early hydrolysis intermediates, the proportions of *myo*-inositol pentakisphosphates produced by the action of wild-type AppA and variant enzymes on phytate can be established (Table 2).

Based on the known stereospecificity of the wild-type enzyme, InsP_5_ [6-OH] accounts for 87% of the InsP_5_ produced with the remaining 13% being InsP_5_ [1/3-OH]. The proton donor variants displayed stereospecificities which varied in relation to the mutation(s) introduced (Figure 4).

Only AppA-HDE showed an increase in stereospecificity, decreasing the production of InsP_5_ [1/3-OH] to only 6% of total InsP_5_. Of the remaining variants, AppA-HAE showed a marginally increased production of InsP_5_ [1/3-OH] (22% of the total InsP_5_), while AppA-HET proved to be more catalytically flexible producing almost equal amounts of InsP_5_ [1/3-OH] and InsP_5_ [4/6-OH]. On the other hand, action of the AppA-HAT variant revealed a reversal of stereospecificity from the wild-type enzyme generating InsP_5_ [1/3-OH] equivalent to 89% of the total InsP_5_s. These differences in the stereospecificity of hydrolysis of InsP_6_ thus resulted in the generation of a range of populations of InsP_5_ species.

Changes in stereospecificity also extended to lower phosphorylated intermediates. For example, AppA-HAE had an increased production of InsP_5_ [1/3-OH] relative to the wild-type enzyme and a change in the profile of InsP_4_s generated (Appendix A) suggesting changes in the stereospecificity of hydrolysis of InsP_5_s. In addition, while AppA-HDE increased specificity towards InsP_6_ relative to the wild-type enzyme, it showed reduced accumulation of 1D-*myo*-Ins(2,3,4,5)P_4_, accumulating instead a novel InsP_2_ intermediate. Therefore, subject to the limitations of our analysis, the position and identity of the catalytic proton donor in AppA has profound effects on the stereospecificity of the enzyme acting not only on phytic acid but also on lower phosphorylated hydrolysis intermediates.

### 2.5. Mutations in the Proton Donor Motif Affects the Size and Polarity of Specificity Pockets

High-resolution X-ray crystallography was employed to provide a structural context for the findings of our solution phase kinetic characterization of AppA variants, and to potentially identify factors influencing stereospecificity. Following the methodology used for wild-type AppA, X-ray crystal structures were therefore solved for complexes of the proton donor motif variants with InsS_6_. These were refined to provide models with low *R*_free_ values at resolutions ranging from 1.41 Å to 2.6 Å (Appendix A, Figure 5).

Comparison of these structures with that of the wild-type enzyme revealed no significant conformational changes in the overall structure had occurred as consequence of the mutations (rmsd vs. AppA-HDT: 0.58 Å for AppA-HET, 0.26 Å for AppA-HAT and -HAE, and 0.24 Å for AppA-HDE). For all the variants, as with the wild-type enzyme, InsS_6_ was modelled to the experimental electron density with the sulfomonoester group at position 1D-6 of the inositol ring pointing towards the catalytic histidine residue, His17. As previously noted, this assignment was unambiguous in the majority of cases due to clear electron density for the unique axial 2-sulfate. However, the electron density for the InsS_6_ molecule bound to the active site of the AppA-HET variant was less well-defined. Given that this variant shows a reduced stereospecificity in hydrolysis of phytate it is possible that the enzyme may be able to bind the substrate analogue in multiple orientations. This phenomenon was previously observed for the MINPP phytase from *B. longum* [40]. Consequently, the sulfomonoester groups at positions 1D-3 and 1D-6 of the inositol ring of InsS_6_ were simultaneously modelled in the primary specificity pocket adjacent to His17 to model static disorder in the binding of the inhibitor (Appendix A). Note that the presence of residual electron density for an axial sulfate group in pockets E and F restricts the orientation of inhibitor binding to one or both possibilities. The inhibitor molecules have refined occupancy ratios 0.47:0.53 (S3:S6) suggesting little intrinsic preference between the two orientations of binding. Certainly, the presence of conformational disorder (i.e., multiple bound conformations) in the InsS_6_ ligand is consistent with the lower positional stereospecificity observed for this variant relative to that seen for the wild-type enzyme and for other canonical HP2 phytases (Barrientos, Scott, and Murthy, 1994; Greiner, Carlsson and Alminger, 2000; Greiner, Alminger and Carlsson, 2001; Ragon et al., 2008). While noting that the enantiomer pairs InsP_5_ [1-OH]/InsP_5_ [3-OH], and InsP_5_ [4-OH]/InsP_5_ [6-OH] cannot be separated by our HPLC approach, our observation of disorder in the binding of a substrate-mimic is consistent with a dual 1D-3/6-positional specificity of the enzyme. In addition, in the structure solved for the AppA-HET variant an inhibitor molecule was also found at the entrance of the active site, just above the inhibitor molecule occupying the catalytic site, and interacting with K24 (Appendix A). Binding at a site immediately adjacent to the catalytic centre, this may represent phytate (or a partially dephosphorylated inositol polyphosphate) in a standby mode between cycles of catalysis. A similar binding mode was observed in the structures of a phytase of the protein tyrosine phosphatase class [45] and also of a wheat purple acid phytase [48].

Analysis of the crystal structures of AppA variants revealed details of the conformations of the sidechains of nominal proton donor residues, and changes in the size and polarity of specificity pockets. Turning first to variants bearing mutations at position 304, substitution of the proton donor residue with an alanine (AppA-HAT) leads to an increase in volume of specificity pockets A and B and a shift of the inhibitor (rmsd 0.5 Å for carbons of the inositol ring) towards pocket B. A modest shift is also observed in the conformations of the sidechains of pocket B residues, His250 and Phe254. This variant shows a marked change in stereospecificity displaying almost exclusive ID-1/3-phytase activity. Indeed, this provides evidence for an intrinsic role played by the proton donor residue itself in determining the stereospecificity of the enzyme. Although we did not determine the absolute enantiomerism of the InsP_5_ product, placing the 1D-1- or 1D-3-phosphate in pocket A results in the axial 2-phosphate occupying pockets B or F, respectively. In the absence of further evidence, the increased size of pocket B following the D304A mutation may point to an enhanced 1-phytase activity of this proton donor-less variant. Why for this variant the inhibitor binds with the 1D-6-sulfate in specificity pocket A rather than the 1D-1- or 1D-3-sulfate as suggested by the results of HPLC analysis is unclear; however, we note that a previous crystallographic study of a variant AppA resulted in a similar phenomenon [11].

The AppA-HET variant, which produces almost equal amounts of InsP_5_ [4/6-OH] and InsP_5_ [1/3-OH], shows the greatest reshaping of the active site. This may help explain the observation of InsS_6_ binding in two conformations of similar energy. The structure reveals the glutamate residue to be found as the *t* rotamer (χ_1_ = 168°) pointing away from the substrate analogue, instead forming a salt bridge with H250 and making only weak interactions with the sulfates bound in pockets A and B. Furthermore, there is no low energy (or otherwise) rotamer described for glutamate which can leave its carboxyl group suitably positioned for to directly participate in catalysis. This is consistent with the low turnover number recorded for this variant. Similar to the AppA-HAT variant, it possesses larger pockets of A and B than the wild-type enzyme because of the absence of the aspartate sidechain. In fact, E304 positions its carboxylic group at the back of pocket B forming a salt bridge with His250 and makes only weak interactions with the sulfates bound in pockets A and B. A change in polarity also occurs in this pocket because of the distal position of the carboxylic acid group of E304. The sum of these changes presumably produces an active site able to accommodate multiple binding modes.

At residue position 305, the T305E substitution found in the AppA-HDE and -HAE variants results in sidechain χ_1_ angles of 46° and 65°, respectively, for the glutamate residue. The residue therefore adopts the *g*^−^ conformation rather than the preferred *g*^+^ conformation for glutamate observed in α-helical environments [49,50,51], which is seen in bacterial MINPPs for which crystal structures are available [25,40]. The *g*^+^ rotamer is required for the carboxyl group of this residue to approach the phosphomonoester bridging oxygen in the specificity pocket A to allow it to function as a proton donor in catalysis. A phenylalanine residue is appropriately positioned in both MINPP structures to preclude the *g*^−^ conformation by steric hindrance; however, no such residue is appropriately positioned to similarly influence E305 in the active site of AppA. Thus, in the AppA-HAE and -HDE variants the E305 sidechain points away from the substrate and towards the active site cavity. The inability of E305 to participate in catalysis is reflected in the similarity of the kinetic parameters for AppA-HAE to those of the proton donor-less AppA-HAT variant and low turnover number for hydrolysis of phytate. AppA-HDE has slightly enlarged pockets A and B due to the loss of the side-chain methyl group of T305, whereas the double mutant AppA-HAE has larger pockets A and B with an overall change in polarity. For both variants, pocket F decreases in size and its polarity increases by virtue of introduction of the larger sidechain of and carboxyl group of glutamate.

### 2.6. Crystal Structures Reveal Snapshots along the Structural Catalytic Cycle

As phytate is rapidly hydrolysed by wild-type AppA, our efforts to capture crystallographic snapshots of its complex with the substrate by soaking crystals with InsP_6_ were unsuccessful. This was the case even at pHs where the enzyme retains minimal activity in solution, yielding only the structure of the orthophosphate (product)-bound complex. The phytase activities of the proton donor motif variants being significantly lower than that of the wild-type enzyme led to trials using crystals of the apo variant enzymes soaked with the highest concentration of phytate possible at the pH of crystal growth (pH 6.5). Whilst these trials again yielded predominantly structures of phosphate-bound enzymes, we were fortunate in being able to solve the structure of the proton donor-less variant, AppA-HAT, at 1.85 Å resolution with the catalytic residue His17 modified as phosphohistidine. Utilizing this structure as a proxy for the enzyme in the covalent intermediate state, the structure of the wild-type enzyme with InsS_6_ as representative of its complex with phytate, and the structure of the complex of the enzyme with orthophosphate as representative of the complex with the second stage product, we can now build on previous descriptions of conformational changes occurring during catalysis [11] to describe snapshots during the complete structural catalytic cycle of the enzyme. This complements the results of a similar study of the MINPP from the gut bacterium, *Bifidobacterium longum* [40].

The active site of the AppA lies between two structural domains: an α- and an α/β-domain. Previous studies [11] have described the binding of InsP_6_ leading to the rearrangement of a region facing the active site involving the loops R20-T26 in the α-domain to allow residues R20, T23, and K24 to make contacts with the substrate. The amino acids displaying the largest displacements are T23 and K24 which close the active site cleft over the substrate. The polypeptide loop W36-W46 makes a smaller conformational change, locking R20-T26 in place in the enzyme–substrate complex. Our structure of AppA in complex with a pseudosubstrate inhibitor confirms these conformational changes and, in addition, being based on the wild-type enzyme allows a full description of the productive enzyme–substrate complex. Figure 6 presents a close-up of the region of the active site involved in the movement.

Following loop closure, the complex is characterized by a number of specific interactions involving catalytic core residues. This catalytic core includes R16 which interacts with the sulfate groups in specificity pockets A and B, and with the sidechain of the proton donor, D304; H17, the catalytic histidine, that lies 2.9 Å from the sulfur atom of the sulfate group in pocket A; R20 that interacts with the sulfate groups in pockets A and B; R92 which interacts with sulfate groups in pockets A, E, and F; and H303 which interacts with sulfate in pocket A. Finally, to complete the catalytic core, the proton donor D304 lies 2.9 Å from the bridging oxygen group of the sulfomonoester bound in pocket A. Beyond the catalytic core residues, T23 and K24 are the only residues to contribute to interactions in specificity pocket D. When acting on lower-phosphorylated inositol polyphosphates such as 1D-*myo*-Ins(2345)P_4_ with the 1D-3-phosphate in pocket A, the absence of a phosphate group in pocket D and consequent loss of interaction with K24 to stabilize the loop following closure may help explain the relatively poor performance of this enzyme towards this and lower intermediates. Indeed, when acting on InsP_3_ and lower intermediates the absence of phosphates in pockets C, D, and E are expected to amplify this effect.

The conformational change is driven by binding of phosphate in pocket A which induces the R20 sidechain to reorient to interact with phosphate bound in specificity pocket A, the catalytic centre. R20 also forms a hydrogen bond to the carbonyl oxygen of A21. This twists the carbonyl of P22 which hydrogen bonds to the amide G45 and leads to the recruitment of the W36-W46 polypeptide via a conformational change. The temperature factors of atoms in both polypeptide loops are reduced in the substrate-bound, closed state. Notably, the involvement of R20 in stabilizing the phosphohistidine intermediate and its interactions with bound orthophosphate in the product complex explain why loop opening does not occur until after loss of the second stage product. As noted previously [11] a further, more subtle conformational change also takes place on substrate binding. In the apo state conformation, the sidechain carboxylate of E219 points out into the lumen of pocket B, making a salt bridge with R16 and interacting with D304 via a water molecule. With phytate bound, unfavourable steric interactions and charge repulsion with the pocket B phosphate reorients this carboxylate group towards D325; the bridged interaction with D304 is lost, a change proposed to enhance the acidity of the proton donor. Access to snapshots of the remaining steps in the structural catalytic cycle allows further insights into the role of E219. Loss of the first stage hydrolysis product and the transition to the phosphohistidine intermediate state reveals a flip of the E219 sidechain carboxylate back to that seen in the initial apo state and presumably an increase in basicity of D304 commensurate with its role as a general base in the breakdown of the phosphohistidine intermediate stabilized by interactions with core residues, particularly H303 and R92. Thus, when pocket B is occupied by a ligand group, E219 is in the inwards looking D304 high acidity state, otherwise it is outward-looking and in the D304 high basicity state. This presumably affects the catalytic efficiency of the enzyme acting on substrates which do not contain adjacent phosphate groups to occupy pockets A and B.

### 2.7. A Conserved Water Molecule Is Positioned to Play Multiple Roles in Catalysis

Conserved water molecules identified in high-resolution crystal structures have been frequently linked to enzyme function [52,53]. In the structure of apo-AppA (PDB entry 1DKL), E219 interacts with the proton donor D304 via a conserved water molecule. That this interaction was lost on phytate binding as E219 swings away from D304 was proposed to signal an involvement of E219 in catalysis by increasing the acidity of the proton donor [11]. We also observed a conformational change in the sidechain of E219 on binding of the pseudosubstrate inhibitor and found this water site conserved in the structure of the complex with orthophosphate. Moreover, we note that in all structures of complexes of AppA variants with InsS_6_, a further conserved water molecule was found positioned between the bridging oxygen of the sulfomonoester bound in pocket A and the main chain nitrogen atom of residue 305 (Figure 5). This water molecule was also found in the structure of the phosphohistidine intermediate. We propose that this water molecule may serve two purposes: Firstly, by interacting with the bridging oxygen of the substrate when bound in specificity pocket A, it improves the leaving group character of the oxygen by increasing the acidity of D304 in the substrate-bound state. Secondly, following loss of the first stage product it is well-positioned to play a role as the source of the catalytic nucleophile involved in the breakdown of the phosphohistidine intermediate. Interestingly, this water occupies the same site as a carboxylic acid group oxygen of glutamic acid of the HAE proton motif in the available crystal structures of bacterial MINPPs [25,40]. That this water site is not observed in the structures of the complex of the H17A variant of AppA with phytate (PDB entries 1DKP and 1DKQ) most likely arises as a consequence of the displacement of the bound substrate relative to the position of the substrate analogue inhibitor of around 0.9 Å due to the loss of the catalytic His17 residue.

## 3. Discussion

The *E. coli* phytase AppA was first reported in 1987 [54] and its catalytic mechanism described in 1992 [20]. Engineered variants have since found successful application as animal feed enzymes [55,56]. AppA has been tested with a number of phosphorylated substrates but shows by far the highest activity towards phytate. However, the rate of hydrolysis of this important nutritional component of animal feeds is reduced for inositol polyphosphate intermediates containing four or fewer phosphate groups [28,29,57]. The proton donor and substrate recognition (RHGxRxh) sequence motifs are highly conserved in the active sites of AppA and other HP2 phytases. Residues acting as proton donors in this class are aspartic acid (via an HDx motif), as in AppA, and glutamic acid (via an HAE motif) [24,32]. To investigate the influence of the identity and position of the putative proton donor residue on the stereospecificity of AppA, the characteristic MINPP HAE proton donor motif was introduced stepwise into the *E. coli* enzyme. Four mutants were generated: the proton donor-less variant D304A (AppA-HAT), variants with glutamate at position 304 (AppA-HET) or 305 (AppA-HDE), and the proton donor swap mutant D326A/T327E (AppA-HAE). The phytase activities and stereospecificities of the wild-type and mutant enzymes were then characterized in solution and, to provide a context for the results, their high-resolution structures solved by X-ray crystallography.

HPLC elution profiles of the *myo*-inositol polyphosphate intermediates produced by action of our panel of variant enzymes on phytic acid show that mutations in the proton donor motif have a pronounced influence on stereospecificity. A marked change is observed for the proton donor-less variant AppA-HAT, which produced a predominant InsP_5_ [1/3-OH] peak, and for AppA-HET which produced InsP_5_ [1/3-OH] and InsP_5_ [4/6-OH] in approximately equal amounts. However, kinetic analysis also reveals that variants with a glutamic acid residue at positions 304 (AppA-HET) or 305 (AppA-HAE) are highly compromised in their ability to participate in catalysis as a proton donor. Both of these observations are neatly explained by consideration of their high-resolution crystal structures. On the contrary, the variant AppA-HDE becomes more stereospecific than the wild type while reducing the accumulation of the 1D-*myo*-Ins(2,3,4,5)P_4_ intermediate. Despite its lower catalytic efficiency, this variant is a good candidate for further investigation into its potential for use in animal feeds. Alterations to catalytic efficiency against lower inositol polyphosphates was also seen for the proton donor swap variant, AppA-HAE, which generated an HPLC elution profile of InsP_5_s similar to the wild-type enzyme; however, it generated a richer array of InsP_4_ species.

Clearly, the substitution of a MINPP-like proton donor motif sequence into AppA does not lead to a simple switch to lower stereospecificity. Overall, the X-ray crystal structures reported herein suggest that, as enzyme activity is sensitive to the position and identity of the proton donor residue, changes in stereospecificity may arise as a consequence of alterations to the volumes and polarities of the active site pockets. In particular, active site comparisons suggest that an enlargement of specificity pocket B allows accommodation of multiple InsS_6_ binding poses (e.g., for AppA-HET), while the decrease in size of pocket F by the substitution T305E (e.g., AppA-HDE) leads to an enhancement of stereospecificity. Further analysis is needed to test these observations. While stereospecificity is undoubtedly the experimental manifestation of a large number of competing molecular influences, introduction of a proton donor residue in the active site of AppA results in a shift to 1D-6-phytase activity. Of the variants investigated, it is clear that an aspartic acid at position 304 is the only arrangement where it can function as an effective catalytic proton donor. However, deployment of a glutamic acid at position 305 permits high activity combined with lower stereospecificity. Thus, our data provide a structural context to the measured kinetic parameters and provides strong support for the view that the position and identity of the proton donor residue is dictated by the fold of the enzyme.

Similar to other characterized phytases, AppA is a distributive enzyme, such that each *myo*-inositol phosphate intermediate dissociates from the enzyme and may act as a substrate in further hydrolysis reactions. With a well-characterized phytate degradation pathway and access to a high-resolution crystal structure, we were interested to see if it were possible to develop rules based on the molecular structure of the enzyme that are consistent with its observed stereochemistry. Phytases of the β-propeller class were structurally characterized and a model based on adjacent catalytic and affinity sites proposed to explain their unusual stereochemistry of hydrolysis of phytate [58]. The first of these sites is responsible for binding and hydrolysis, while the second serves to increase the affinity and activity towards phytate or for lower inositol polyphosphates which contain two adjacent phosphate groups. AppA has six identifiable specificity pockets. Pocket A is responsible for binding and hydrolysis of phosphate while the remainder are populated to a greater or lesser extent by active site residues (Appendix A). Specificity pocket B, containing nine residues within 5 Å of the bound sulfate group of InsS_6_ is the best defined, while pocket D is the least, having only two residues (T23 and K24) which are recruited as a result of the conformational change which occurs on substrate binding. On the basis of enumeration of contacts with phosphate groups of the substrate and by analogy with the β-propeller phytases, we propose that specificity pocket B constitutes the primary affinity site for the enzyme, followed by pockets C and F.

Bearing in mind the stereochemical restraints imposed by the active site, we can predict in general terms the nature of the binding of the phosphate groups of phytate and its lower phosphorylated hydrolysis products to the active site specificity pockets (Figure 7).

Consideration of this leaves us with a simple set of rules based on our crystal structures which adds to our understanding of the structural basis for the sequential cleavage of phytate by AppA. Firstly, for high catalytic activity the scissile phosphate must be accompanied by inserting an adjacent phosphate on the inositol ring into the primary affinity site, specificity pocket B. This is presumably due to the high specificity that can be achieved but also because by doing so the acidity of D304 can be enhanced through conformational change involving E219. It is also in keeping with reports that HP2 phytases prefer dephosphorylation of adjacent phosphate residues [57]. Secondly, higher affinity can be achieved by docking further phosphates into specificity pockets C and F. Thirdly, occupation of specificity pocket D is essential for efficient catalysis. However, occupation of pocket D by the 2-phosphate is excluded on stereochemical grounds as the axial phosphate group is directed away from T23 and K24 and unable to engage with them, presumably leading to lower affinity binding. These three simple rules are consistent with the observed 6/1/3/4/5 sequential hydrolysis of phosphate groups of phytate by AppA. The enzyme’s Achilles’ heel, the relatively poor efficiency of hydrolysis of InsP_4_ and lower inositol polyphosphates, presumably arises from the existence of the mobile loop in the α-domain which provides the only two residues of pocket D. Loop closure, existing presumably to promote substrate access and product egress, leaves the enzyme poorly equipped to hydrolyse substrates lacking an equatorial phosphate in pocket D, thus precluding full loop closure and efficient catalysis. That other HP2 1D-6-phytases do not undergo conformational change in substrate binding led to the suggestion that they preformed active sites tailored for activity towards phytate but with lower specific activities [13,14].

Complete dephosphorylation of phytate may have advantages to the developing animal other than that resulting simply from enhancing bioavailable phosphate. Indeed, inositol has been shown to be an important regulator of the transport and deposition of fat, leading to the suggestion that it may help support animal growth [59]. Given the importance of animal feed enzymes to global food security, it is hoped that the results of this study may be of use to those seeking to develop less stereospecific enzymes which can alone, or in combination with other phytases, degrade InsP_6_ to completion. Indeed, progress in this direction is already forthcoming from a directed evolution approach [31]. Other approaches might usefully include investigations of the effects on stereochemistry of an exchange of other active site elements between MINPPs and AppA. A similar strategy was successfully employed to transfer favourable catalytic properties from *Aspergillus niger* NRRL 3135 phytase to a synthetic phytase (phytase-1) by swapping active sites [60]. Finally, given its role as the primary affinity site in AppA, saturation mutagenesis of residues in specificity pocket B may prove a fruitful route to the discovery of next generation animal feed phytases.

## 4. Materials and Methods

### 4.1. Cloning and Site-Directed Mutagenesis

The gene encoding the *Escherichia coli* periplasmic phytase, AppA, was amplified from the genome of strain BL21(DE3) pLysS (Invitrogen, Waltham, MA, USA) and cloned into the expression vector pOPINB [61]. pOPINB was a gift from Ray Owens (Addgene plasmid # 41142; http://n2t.net/addgene:41142; accessed on 1 January 2019). The construct was designed for the isopropylthio-β-D-galactoside (IPTG) inducible expression of an N-terminal cleavable His-tag recombinant protein in the cytoplasm of a suitable *E. coli* host strain. The sequence was cloned in a truncated form lacking the periplasmic signal peptide (residues 1–22) and fused to an N-terminal 3C-protease cleavable His_6_-tag (residues MAHHHHHHSSGLEVLFQ|GP, where | indicates the 3C-protease cleavage site). The sequence of this construct (named 6H3C-AppA) was confirmed through nucleotide sequencing. Site-directed variants were generated from this construct using the method reported by Liu and Naismith [62]. Note that the residue numbering used in this report refers to the amino acid sequence of the mature enzyme lacking the signal peptide, i.e., the catalytic histidine residue becomes H17 and the proton donor residue, H304.

### 4.2. Protein Expression and Purification

Expression of 6H3C-AppA and its variants was performed using the *E. coli* strain Shuffle Express T7 (New England Biolabs). Following transformation, cultures were incubated at 37 °C and 180 rpm. On reaching an OD_600_ of 0.9 induction was performed with 0.01 mM IPTG at 37 °C. Cells were left to grow overnight (o/n) following which they were harvested by centrifugation at 4 °C, 5500 rpm for 20 min. Pellets were resuspended in 30 mL Tris-Cl lysis buffer containing 20 mM Tris pH 8.0, 300 mM NaCl, 10 mM imidazole, 0.5 % Triton X-100, and snap frozen in liquid nitrogen and stored at −80 °C before thawing. Cells were lysed by means of a French pressure cell press. The soluble fraction of the sample was separated by centrifugation at 4 °C, 40,000× *g* for 1 h the overexpressed protein was isolated by Ni-NTA immobilized metal chelate chromatography (IMAC) over a 10–1000 mM imidazole gradient at pH.8.0. Cleavage of the His-tag was carried out by adding His-tagged 3C-protease at a concentration 40× lower than that of the overexpressed protein, followed by overnight dialysis against a cleavage buffer containing 50 mM Tris-HCl pH 7.4, 500 mM NaCl, and 2.5 mM CaCl_2_. A second Ni-NTA IMAC step was carried out to separate detagged AppA from residual tagged protein and 3C protease. Cleaved AppA was concentrated using an Amicon Ultra centrifuge filter unit (10 kDa cut-off) and gel filtered using a HiLoad 16/600 Superdex 75 pg size exclusion column (GE Healthcare) and a running buffer containing 20 mM HEPES pH 7.4 and 150 mM NaCl. Protein fractions with a purity of at least 99% as estimated by SDS-PAGE were pooled and concentrated as performed previously. Estimates of the enzyme concentrations were made from absorbance measurements at 280 nm using a NanoDrop One Microvolume UV Spectrophotometer (Thermoscientific, Waltham, MA, USA).

### 4.3. Phosphate Release Assay

This assay allows the determination of the free phosphate released by hydrolysis of InsP_6_ by the molybdenum blue reaction. The absorbance of molybdenum blue is measured at 700 nm and is proportional to the orthophosphate concentration. A typical calibration curve showed the assay to be linear in the range from 10 μM to 2.5 mM orthophosphate. A total of 5 mM phytic acid dipotassium salt (≥95 % pure) (Sigma-Aldrich Inc., St. Louis, MO, USA) was used as a substrate. Reactions were performed in triplicate at room temperature and pH 5.5 unless otherwise stated. Reactions of 50 or 100 µL volume were stopped by the addition of equal amounts of a freshly prepared solution made of four parts of reagent A (12 mM ammonium molybdate tetrahydrate, 5.4% saturated sulfuric acid) and one part of reagent B (0.4 M iron(II) sulfate heptahydrate plus a few drops of saturated sulfuric acid). Absorbance was measured at 700 nm after 30 min using a Hidex Sense plate reader. Control reactions of buffer only, substrate only, and enzyme only were set up simultaneously as well as a calibration curve of increasing concentration of orthophosphate.

### 4.4. Determination of Enzyme Kinetic Parameters

Reactions of total volume 100 μL were set up in triplicate at fixed concentrations of enzymes (wild-type AppA and the HDE variant at a concentration of 7.5 nM, and all other variants at 1 μM). *myo*-inositol 1,2,3,4,5,6-hexakisphosphate, and dodecasodium salt (*Zea mays*, 99% pure, Merck) was used as substrate at concentrations 25, 50, 100, 200, 400, 600, 800, and 1200 μM. Reactions were incubated at room temperature and allowed to proceed for 5, 10, 15, 20, 25, and 30 min. The buffer chosen was 200 mM sodium acetate pH 4.5, 0.15 M NaCl. Reactions were inactivated by addition of molybdenum blue reagent in equal part and the absorbance at 700 nm was measured after 30 min incubation of the samples with the stopping reagent. Data were processed with the ‘nls’ function provided in R (see https://stat.ethz.ch/R-manual/R-devel/library/stats/html/nls.html; accessed on 1 January 2021) that determines the nonlinear least-squares estimates of the parameters of a nonlinear model. In this analysis, the non-linear model is the Michaelis–Menten equation. The goodness of fit of the model was confirmed by checking residual error values and *t*-test.

### 4.5. Inhibition of Phytase Activity by Myo-Inositol Hexakissulfate

Reactions of total volume 50 μL were set up in triplicate at fixed concentrations of enzyme (7.5 nM). A total of 1 mM phytic acid dipotassium salt (≥95 % pure) (Sigma-Aldrich Inc., St. Louis, MO, USA) was used as substrate. Serial dilutions of *myo*-inositol hexakissulfate (InsS_6_) (Alfa) at concentrations: 0 nM, 1 nM, 10 nM, 100 nM, 1 μM, 10 μM, 100 μM, and 1 mM were mixed to two buffered solutions of pH 3.0 (0.2 M Glycine-Cl, 0.15 M NaCl) and pH 5.5 (0.2 M sodium acetate, 0.15 M NaCl). Reactions are incubated at 37 °C for 30 min before inactivation by addition of molybdenum blue reagent and absorbance at 700 nm was measured after 30 min incubation with the stopping solution.

### 4.6. Identification of Inositol Polyphosphates by HPLC

Enzymes were used at 25 nM in 200 mM sodium acetate pH 5.5, 150 mM NaCl. *myo*-inositol 1,2,3,4,5,6-hexakisphosphate, and dodecasodium salt (InsP_6_, 1 mM, *Zea mays*, 99% pure, Merck) was used as a substrate. Reactions were stopped at 5, 10, 20, 30, 45, 60, and 120 min by boiling samples at 100 °C for 10 min. Samples were diluted 5× before injection. Inositol polyphosphate standards were generated by the hydrolysis of InsP_6_ in 1 M HCl, 120 °C for 24 h. The HPLC system consisted of a first pump for sample injection (Jasco PU-2089 I Plus—Quaternary inert Pump) connected in series to two CarboPAC PA200 columns (3 × 50 mm, 3 × 250 mm) in which InsP_x_ species were efficiently separated (enantiomers however cannot be resolved) before reaching a reaction coil in which they were chaotically mixed with a reagent (0.1 % Fe(NO_3_)_2_, 2% HClO_4_), which was injected by a second pump (Jasco PU-1585 Intelligent HPLC Pump). This allows UV absorbance detection at 290 nm (range 1.28 nm, Jasco UV 1575 Intelligent UV/Vis detector—16 µL cell). Samples were separated in a methane sulfonic acid gradient (0–0.6 M), flow rate 0.4 mL/min, with water as a counter eluent, reagents were added post-column at a flow rate of 0.2 mL/min. The total run time for each sample was 50 min: 25 min of gradient, 14 min of 0.6 M methane sulfonic acid, and 11 min of water. The peak areas were calculated by integration using the software provided by Jasco (ChromNAV, version 1.19.01). The identities of inositol polyphosphates generated during hydrolysis were determined by reference to the retention times of peaks resulting from a standard sample of chemically hydrolysed InsP_6_ (HCl, 120 °C, 24 h).

### 4.7. Protein Crystallization

Purified wild-type AppA and its HDT-motif variants were concentrated to 4 mg/mL in 200 mM sodium acetate buffer pH 4.5. The proteins were crystallized by sitting drop vapour diffusion. Initial crystallization conditions explored were those reported by Wu et al. [63] (100 mM HEPES pH 7.5, 20% PEG 3350, 2% PEG 8000, 10 mM NiCl_2_) using 1 µl drops incubated at 25 °C. The crystals obtained, however, were of a very thin needle habit, fragile, and only diffracted to around 2.0 Å resolution. In attempts to improve the limiting resolution of diffraction, alternative crystallization conditions were sought using sitting drop vapour diffusion crystallization experiments and the commercial crystallization screens, Structure Screen I/II, and JCSG Plus (Molecular Dimensions Ltd., Newmarket, UK). These experiments used a purified enzyme preparation concentrated to 1.6 mg/mL in a buffer containing 50 mM Tris-HCl pH 7.5, 0.15 M NaCl. Crystallization plates were incubated at 25 °C. Crystals of apo-protein grew in multiple conditions and morphologies. The best crystals were of a rod habit in space group P2_1_ grown in 100 mM MES pH 6.5, 18% *w*/*v* PEG 8000, and 200 mM calcium acetate and diffracted to a maximum resolution of 1.37 Å. The four AppA variant enzymes also crystallized in these conditions. Attempts were made to soak these crystals with InsP_6_ or 1D-*myo*-Ins(2,3,4,5)P_4_ at pH 6.5 but these experiments revealed only structures for the phosphate-bound enzyme, except in the case of the AppA-HAT variant soaked with phytate. Soaking experiments at higher pH (where the enzymes are less active) were also unsuccessful. Complexes of the enzymes with the inhibitor InsS_6_ (*myo*-inositol hexakissulfate hexapotassium salt; Alfa Chemistry) were prepared by soaks of apo-protein crystals at low pH (pH 3.0). This was the pH at which the strongest enzyme inhibition was recorded. Initial attempts at cocrystallization were unsuccessful due to the low binding affinity for InsS_6_ at pH 6.5. The optimal soak condition was subsequently found to be 20% *w*/*v* PEG 3350, 9 mM InsS_6_ in 0.6 M glycine pH 3.0, and 20 % glycerol for 10–25 min at 16 °C. This condition limited pH changes during the soak by employing an increased buffer concentration. To prepare these soaks, InsS_6_ was maintained at its maximum soluble concentration by diluting a 30 mM stock solution preincubated at 42 °C. Soaked crystals were then transferred to a cryoprotectant solution containing 22% *w*/*v* PEG 3350, 9 mM InsS_6_, 0.6 M glycine pH 3.0, and 30% *v*/*v* glycerol, harvested in litholoops (Molecular Dimensions Ltd., Newmarket, UK) and stored in liquid nitrogen.

### 4.8. X-ray Diffraction Data Collection and Crystal Structure Determination

Diffraction experiments were performed at the beamlines I03 and I04 of the Diamond Light Source (Oxfordshire, UK). Data reduction was performed with xia2 [64]. Initial phases for all crystal structures were determined by molecular replacement using the program Phaser [65] and the structure an AppA mutant (PDB entry 1DKL) [11]. Convergence of cycles of rebuilding in Coot [66] and refinement using phenix.refine [67], [68] provided refined structures with the single copy of the enzyme in the asymmetric unit (ASU). Calculation of Polder OMIT maps [69] revealed significant residual electron density in active sites of the ASU corresponding to either bound orthophosphate or InsS_6_. In the case of the AppA-HET variant complex with InsS_6_, careful inspection revealed static disorder and the inhibitor was added to the model in two orientations presenting either the D-6 or D-3-sulfate bound at the catalytic centre. Moreover, inspection of OMIT maps for the apo AppA-HAT variant using diffraction data collected from a crystal soaked with phytic acid showed residual electron density interpreted as a phosphohistidine-modified catalytic residue, His17. Refinement (including occupancy refinement any bound ligands) yielded final structural models which were validated using MolProbity [70] and the wwPDB Validation Service (https://validate.wwpdb.org; accessed on 1 February 2022). Data collection and refinement statistics for all structures are reported in Appendix A. Structural comparisons were carried out using DynDom [71]. Figures were generated using PyMOL version 2.5 [72].

## Figures and Tables

**Figure 1 ijms-23-06346-f001:**
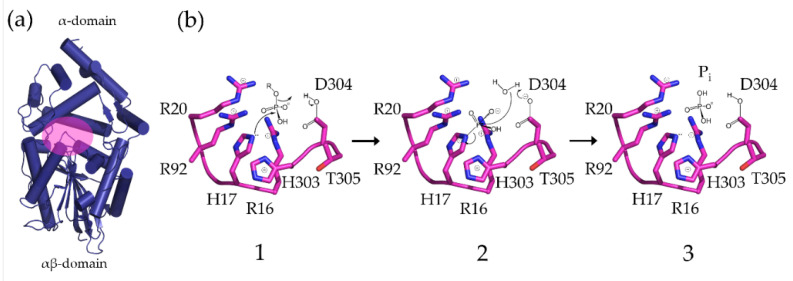
The domain structure and catalytic mechanism of AppA. (**a**) A cartoon representation of the structure of *Escherichia coli* periplasmic phytase (AppA) (α-helices shown as cylinders and β-sheet as arrows). AppA folds to provide an α-domain (upper) and an α/β-domain (lower). The active site cleft (region indicated in pink) is found at their interface. (**b**) Steps in the catalytic mechanism. Core active site residues including selected residues of the nucleophilic (RHGxR, residues 16–20) and proton donor (HDT, residues 303-305) motifs are shown. (1) Nucleophilic attack by H17 to form phosphohistidine intermediate. D304 acts as general acid leading to release of the lower phosphorylated inositol; (2) D304 catalyses breakdown of the phosphohistidine intermediate by acting as a general base; and (3) orthophosphate-bound state. Note that regeneration of the enzyme can occur by donation of the inorganic phosphate group to acceptors other than water.

**Figure 2 ijms-23-06346-f002:**
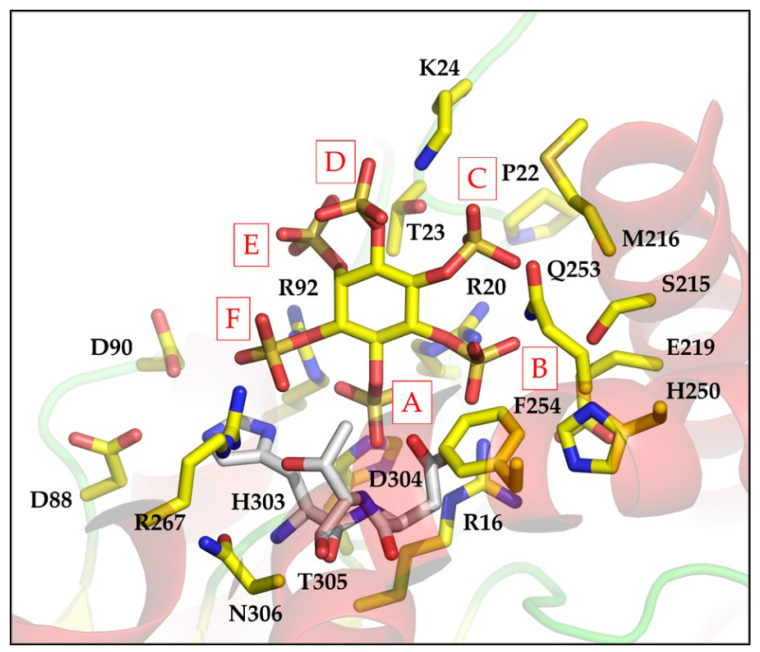
Crystal structure of the complex of AppA with InsS_6_ defines active site specificity pockets. Sidechains of residues in the active site of AppA within 6 Å of an atom of bound InsS_6_ are shown as sticks. Carbon atoms of all residues are coloured yellow except for those of the HDT proton donor sequence motif (residues 303–305) which are coloured grey. Nitrogen atoms are coloured blue and oxygen red. The sulfur atoms of the sulfate groups of the inhibitor are coloured bronze. Overlayed is a semi-transparent rendering of the cartoon secondary structure of the enzyme. Active site specificity pockets are labelled in red capital letters (**A**–**F**).

**Figure 3 ijms-23-06346-f003:**
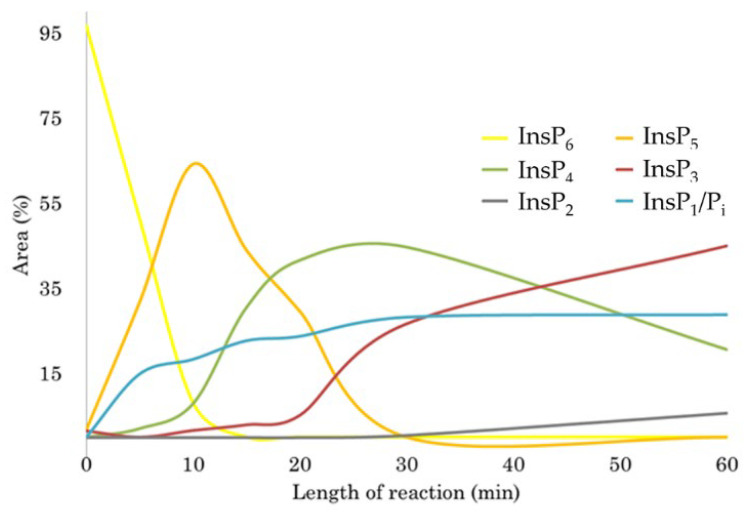
Accumulation of inositol polyphosphate intermediates during the hydrolysis of InsP_6_ by AppA.

**Figure 4 ijms-23-06346-f004:**
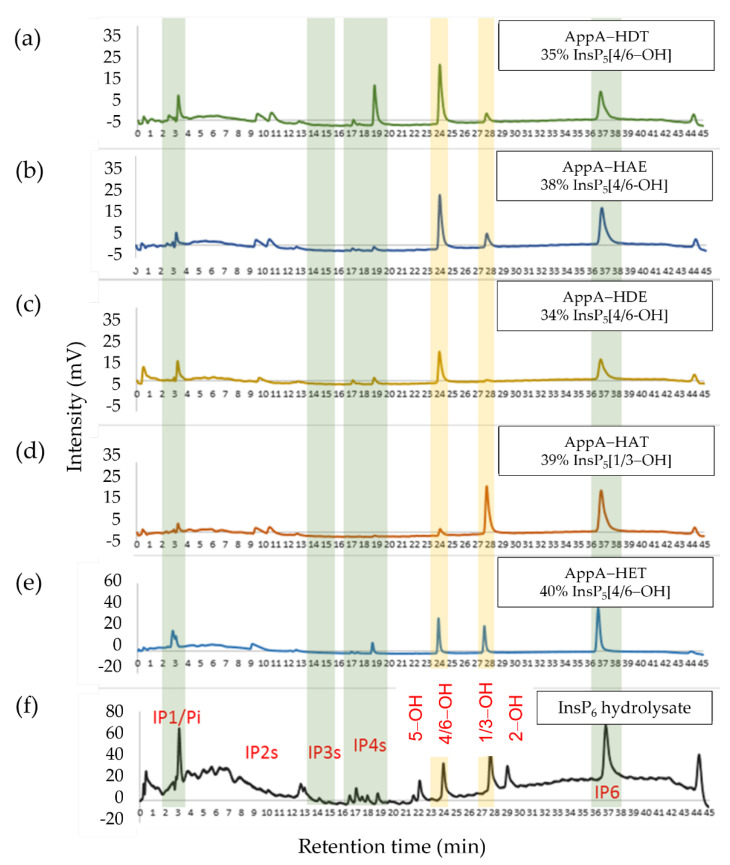
HPLC chromatograms of the products of initial hydrolysis of InsP_6_ by wild-type AppA and proton donor motif variants. Reactions stopped when total InsP_5_ peak area was equal to 34–40% of total. The enzyme proton donor motif sequence and the predominant InsP_5_ peak area (%) are reported on the top right corner of each chromatogram. (**a**) AppA-HDT (wild type), (**b**) AppA-HAE, (**c**) AppA-HDE, (**d**) AppA-HAT, (**e**) AppA-HET, and (**f**) chromatogram of an acid hydrolysate of the substrate (InsP_x_ standards) are shown for reference. The characteristic elution volume ranges for the various inositol polyphosphates are highlighted by vertical-coloured backgrounds. The nomenclature used to identify inositol polyphosphate intermediates is simplified for the purposes of clarity.

**Figure 5 ijms-23-06346-f005:**
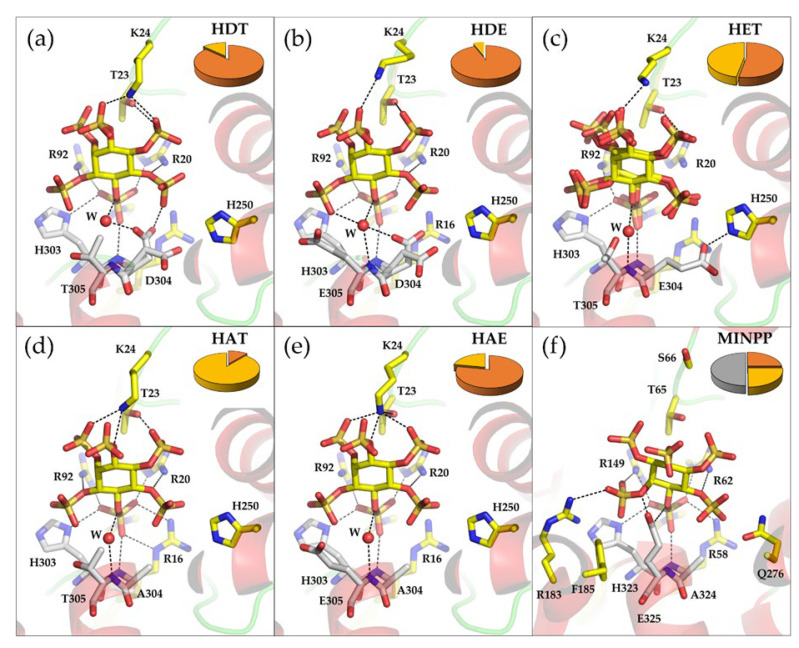
Crystal structures of complexes of AppA proton donor motif variants with the non-hydrolysable substrate analogue, InsS_6_. (**a**) AppA-HDT (wild type), (**b**) AppA-HDE, (**c**) AppA-HET, (**d**) AppA-HAT, (**e**) AppA-HAE. For comparison, panel (**f**) shows the active site of the MINPP from *Bacteroides thetaiotaomicron* (BtMINPP; PDB entry 4FDU). BtMINPP has a characteristic HAE proton donor sequence motif. In all panels, active site residues and bound InsS_6_ inhibitor are shown in stick format with oxygen atoms coloured red, nitrogen blue and sulfur orange. Carbon atoms are coloured yellow except for the residues of the proton donor motif where they are coloured grey. Active site residues are labelled as is a conserved water molecule (W) found within hydrogen bonding distance of the bridging oxygen atom to the sulfomonoester group bound in specificity pocket A. Hydrogen bond interactions of 3.0 Å or less are indicated by black dashed lines. Enzyme secondary structure is shown as semi-transparent cartoon with α-helix coloured red and coil green. The pie charts in panels (**a**–**f**) show the proportions of InsP_5_ [4/6-OH] (orange), InsP_5_ [1/3-OH] (yellow), and InsP_5_ [5-OH] (grey) generated by action of that variant or MINPP on InsP_6_.

**Figure 6 ijms-23-06346-f006:**
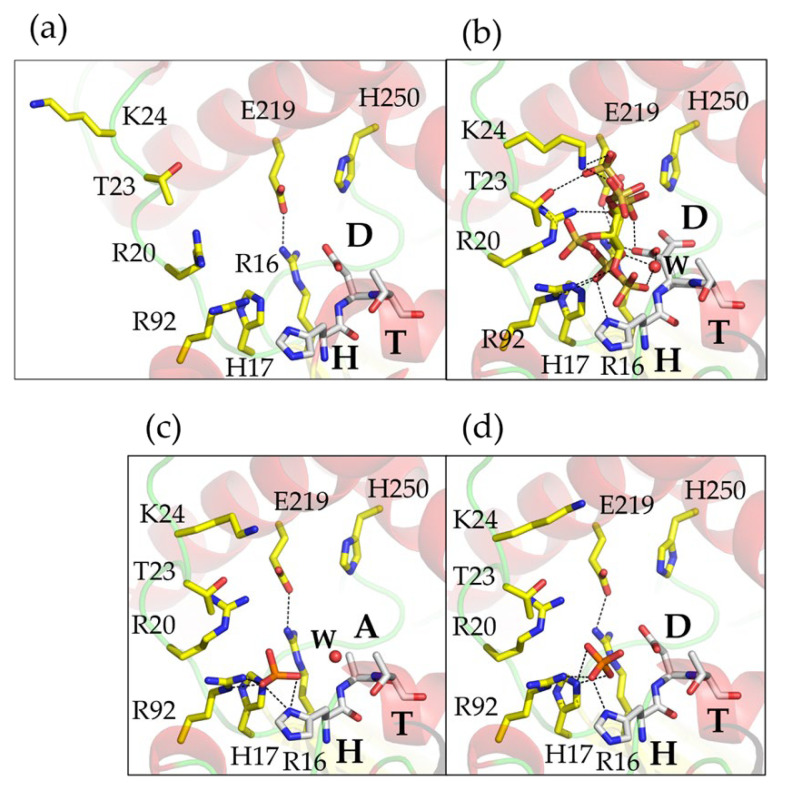
Snapshots along the structural catalytic cycle of AppA. Panels show (**a**) apo-enzyme (apo-enzyme; PDB entry 1DKL), (**b**) AppA complex with InsS_6_ (model for the E.S complex; this study; PDB entry 7Z2S), (**c**) AppA-HAT phosphohistidine intermediate (model for the catalytic phosphohistidine intermediate; this study; PDB entry 7Z32), and (**d**) AppA complex with orthophosphate (product complex; this study; PDB entry 7Z1J). In all panels, active site residues are shown in stick format with carbon atoms coloured yellow (except the residues of the proton donor motif where they are coloured grey), oxygen red, nitrogen blue, and phosphorus dark orange. The complex with InsS_6_ shown in panel (**b**) can also be seen from another orientation in Figure 5a. Hydrogen bond interactions of 3.0 Å or less are indicated by black dashed lines. Enzyme secondary structure is shown as semi-transparent cartoon with α-helix coloured red and coil green. Active site residues are labelled individually except for those of the proton donor motif which are indicated by their single letter code. Further labelled is the conserved water molecule (W) found within hydrogen bonding distance of the bridging oxygen atom to the sulfomonoester group bound in specificity pocket A (see also Figure 5a–e). This water site is also occupied in the structure of the phosphohistidine intermediate shown in panel (**c**).

**Figure 7 ijms-23-06346-f007:**
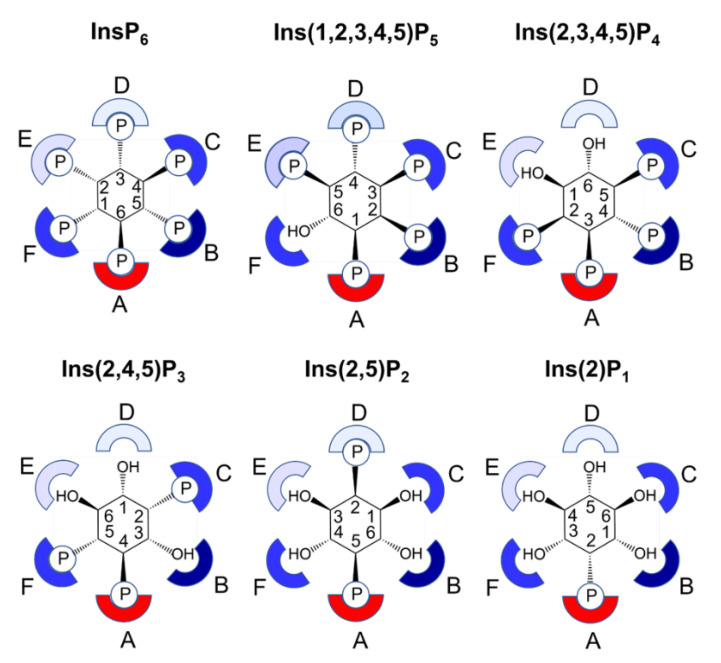
Predicted binding of InsP_6_ and its lower phosphorylated derivatives to the active site specificity pockets of AppA. Specificity pockets are represented by sockets either coloured red (indicating the catalytic pocket) or a shade of blue according to the number of active site residues predicted to lie within 5 Å of a bound phosphate group of the substrate (dark blue-most residues, light blue-least residues). The *myo*-inositol ring of the substrate is shown as a hexagon with carbon atoms numbered by the 1D-notation. Phosphate groups are indicated by a capital P inside a circle with locant, assigned by 1D-notation, of the corresponding inositol carbon position shown.

**Table 1 ijms-23-06346-t001:** Kinetic parameters for wild-type AppA and its variants. Standard errors shown in brackets.

AppA Variant	*K*_m_ (µM)	*k*_cat_ (s^−1^)	*k*_cat_/*K*_m_ (µM^−1^s^−1^)
HDT	161 (3)	10,209 (421)	64 (9)
HAT	32 (7)	10.5 (0.4)	0.32 (0.07)
HET	171 (28)	128 (6)	0.7 (0.1)
HAE	33 (4)	12.5 (0.3)	0.38 (0.05)
HDE	92 (5)	1689 (24)	18 (1)

**Table 2 ijms-23-06346-t002:** Proportions of *myo*-inositol pentaphosphates produced by the action of wild-type AppA and proton donor motif (PD) variant enzymes on phytate.

PD Motif Variant	InsP_5_ [4/6-OH] (%)	InsP_5_ [1/3-OH] (%)
HDT	87	13
HDE	94	6
HAE	78	22
HET	53	47
HAT	11	89

## Data Availability

Coordinates and diffraction data for the crystal structures of AppA in the phosphate bound (PDB entry 7Z1J) and InsS_6_ bound (7Z2S) forms, of the AppA HAT-mutant in the phosphohistidine intermediate (7Z32) and InsS_6_ bound (7Z2T) forms, and of the AppA-HET, -HAE and -HDE mutants in complex with InsS_6_ (7Z3V, 7Z2W and 7Z2Y, respectively) have been deposited in the PDB.

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
