# Peer review of "Insights to the Structural Basis for the Stereospecificity of the Escherichia coli Phytase, AppA"

_ijms, 2022, doi:10.3390/ijms23116346_

Round 1
Reviewer 1 Report
The authors have investigated the influence of the proton donor residue on the stereospecificity of AppA, and introduced the characteristic MINPP HAE proton donor motif into the E. coli enzyme. The four mutants were generated: AppA-HAT, AppA-HET, AppA-HDE, and AppA-HAE. The phytase activities and stereospecificities were examined in solution and the X-ray structures were determined.
The experiments are well-performed, and the manuscript is well written. The HAT mutant showed that the reversal of the stereospecificity shown in the Table 2 and Figure 4, but the crystal structures of the wild type and HAT mutant showed the same binding of the InsS6. If there is any plausible explanation, the authors should clearly mention the origin of the reversal stereospecificity of the HAT mutant based on the X-ray crystallographic structure.
There are many typographical errors. The authors should check all the text.
line 31, “targetting” should be “targeting”.
line 124, “of of ” should be “of”.
line 147, “afffinity” should be “affinity”.
line 152 and 264, “identication” should be “identification”.
line 391, “stereospecificty” should be “stereospecificity”.
line 433, “snaphots” should be “snapshots”.
line 592, “stong” should be “strong”.
line 762-769 “crystallisation” should be “crystallization”.
Author Response
We thank the reviewer for their careful and critical reading of the manuscript. We have corrected the typographical errors and added the sentence copied below to address the unexpected binding of the pseudosubstrate inhibitor to the AppA-HAT variant (lines 396-399):
Why for this variant the inhibitor binds with the 1D-6-sulfate in the A specificity pocket rather than the 1D-1- or 1D-3-sulfate as suggested by the results of HPLC analysis is unclear, however, we note that a previous crystallographic study of a variant AppA resulted in a similar phenomenon [11].
Author Response
Following the suggestion of the reviewer, the resolution ranges shown in Table S1 of the Supplementary Information have been modified to show two decimal places for high resolution limits and only one for low resolution.